# Resveratrol Improves Endothelial Function by A PREP1-Mediated Pathway in Mouse Aortic Endothelial Cells

**DOI:** 10.3390/ijms241511891

**Published:** 2023-07-25

**Authors:** Serena Cabaro, Ayewa L. Agognon, Cecilia Nigro, Sonia Orso, Immacolata Prevenzano, Alessia Leone, Cristina Morelli, Federica Mormone, Serena Romano, Claudia Miele, Francesco Beguinot, Pietro Formisano, Francesco Oriente

**Affiliations:** 1Department of Translational Medicine, Federico II University of Naples and URT Genomic of Diabetes of Institute of Experimental Endocrinology and Oncology, National Council of Research (CNR), Via Pansini 5, 80131 Naples, Italy; serena.cabaro@unina.it (S.C.); ayewa.agognon@gmail.com (A.L.A.); c.nigro@ieos.cnr.it (C.N.); sonia.orso@hotmail.com (S.O.); imma987@hotmail.it (I.P.); aleleone86@hotmail.com (A.L.); cristinamorelli@tim.it (C.M.); federica.mormone@outlook.it (F.M.); serena.romano2@unina.it (S.R.); c.miele@ieos.cnr.it (C.M.); beguino@unina.it (F.B.); foriente@unina.it (F.O.); 2Center for Basic and Clinical Immunology Research (CISI), University of Naples Federico II, 80131 Naples, Italy

**Keywords:** PREP1, endothelial cells, resveratrol, eNOS, inflammatory cytokines, antioxidant molecules, *Prep1* hypomorphic heterozygous mouse model

## Abstract

PREP1 is a homeodomain transcription factor that impairs metabolism and is involved in age-related aortic thickening. In this study, we evaluated the role of PREP1 on endothelial function. Mouse Aortic Endothelial Cells (MAECs) transiently transfected with a *Prep1* cDNA showed a 1.5- and 1.6-fold increase in eNOS^Thr495^ and PKCα phosphorylation, respectively. Proinflammatory cytokines *Tnf-α* and *Il-6* increased by 3.5 and 2.3-fold, respectively, in the presence of *Prep1*, while the antioxidant genes *Sod2* and *Atf4* were significantly reduced. Bisindolylmaleimide reverted the effects induced by PREP1, suggesting PKCα to be a mediator of PREP1 action. Interestingly, resveratrol, a phenolic micronutrient compound, reduced the PREP1 levels, eNOS^Thr495^, PKCα phosphorylation, and proinflammatory cytokines and increased *Sod2* and *Atf4* mRNA levels. The experiments performed on the aorta of 18-month-old *Prep1* hypomorphic heterozygous mice (*Prep1^i/+^*) expressing low levels of this protein showed a 54 and 60% decrease in PKCα and eNOS^Thr495^ phosphorylation and a 45% reduction in *Tnf-α* levels, with no change in *Il-6*, compared to same-age WT mice. However, a significant decrease in *Sod2* and *Atf4* was observed in *Prep1^i/+^* old mice, indicating the lack of age-induced antioxidant response. These results suggest that *Prep1* deficiency partially improved the endothelial function in aged mice and suggested PREP1 as a novel target of resveratrol.

## 1. Introduction

Insulin resistance and biological aging are characterized by increased inflammation and reactive oxygen species (ROS) generation, which can predispose individuals to vascular injury. A reduction in nitric oxide (NO) production and the impairment of endothelial nitric oxide synthase (eNOS) activity can also contribute to endothelial dysfunction [1,2]. eNOS phosphorylation represents one of the main mechanisms regulating enzymatic activity. In particular, phosphorylation on Ser^1177^ has been shown to have an activating function and can be induced by several protein kinases, including Akt/PKB. On the other hand, phosphorylation on Thr^495^ has an inhibitory action and is promoted by protein kinase C (PKC) or AMP-activated protein kinase (AMPK) [3,4,5,6,7,8,9].

In recent years, several dietary supplements and nutraceuticals have been used in the prevention and/or treatment of aging, including metabolic and cardiovascular diseases (CVD) [10,11,12]. In particular, resveratrol (3,5,4′-trihydroxystilbene) (RSV), a polyphenolic phytoalexin found in grapes, blueberries, and red wine, has been shown to have numerous health benefits on cardiovascular diseases, cancer, diabetes, inflammation, neurodegeneration, and aging. At the molecular level, RSV, by activating protein kinases, can regulate different transcription factors.

The RSV-mediated tumor suppressive function has been associated with an upregulation in the activating transcription factor 2 (Atf2), while other evidence has indicated that RSV, by increasing the early growth response factor 1 (EGR1), induces proliferation arrest and programmed cell death in lung cancer cells [10,11]. Interestingly, RSV improves brain function in aged mice through the transcriptional activation of the cyclic AMP-response element (CRE)-binding protein (CREB) [12]. In addition, other researchers have suggested that RSV exerts a powerful antioxidant response through the activation of the NF-E2-related factor-2 (Nrf2) transcription factor, leading to an improvement in age-related inflammation, neurodegeneration, and cardiovascular dysfunction [10].

RSV also plays an important role in endothelial function and glucose metabolism. In human umbilical vein endothelial cells (HUVECs), RSV attenuates endothelial oxidative injury by inducing autophagy in a transcription factor EB (TFEB)-dependent manner [13]. Moreover, RSV, by attenuating the expression of members of the class O of forkhead box transcription factors (FOXO) 1 and 3a, has been shown to ameliorate adipose tissue insulin resistance in type 2 diabetic rats [14].

We and others have identified a member of three-amino acid loop extension (TALE) homeodomain transcription factors named PREP1, which impairs both glucose and lipid homeostasis in insulin-target tissues, such as muscle, adipose tissue, and the liver [15,16,17,18,19]. Most of these results have been obtained in vivo using both hypomorphic homozygous (*Prep1^i/i^*) and hypomorphic heterozygous (*Prep1^i/+^*) mouse models, expressing 3–7 and 55–57% of the protein, respectively. Adult mice lacking the *Prep1* feature with better peripheral glucose and lipid metabolism were found to not develop diabetes and steatohepatitis in the presence of a streptozotocin and methionine-choline deficient diet, respectively [18,19]. PREP1 also plays an important role at the vascular level. Indeed, PREP1 increases several pro-angiogenic factors, including some inflammatory cytokines such as *Il-6* and *Tnf-α,* and stimulates tube formation in murine aortic endothelial cells (MAECs). In parallel, *Prep1^i/+^* mice show attenuated placental vessel formation [20]. More recently, we have observed that PREP1, by inducing aortic wall thickening, can contribute to vascular dysfunction with advancing age. In particular, an increase in PREP1 has been found in the aorta of aged mice and in senescent vascular smooth muscle cells (VSMCs), which promotes neointimal formation by reducing VSMCs apoptosis [21]. In the present work, we evaluated the role of PREP1 in endothelial function. Our results indicate that *Prep1* overexpression in MAECs reduces eNOS activation and expression in antioxidant genes and increases proinflammatory cytokines through a PKCα-dependent pathway. Resveratrol, by regulating PREP1, reverts these effects. Finally, aged *Prep1^i/+^* mice showed increased aortic eNOS activation and reduced *Tnf-α* expression, while antioxidant genes decreased compared to same-age WT littermates, despite the lower PREP1 levels.

## 2. Results

### 2.1. Evaluation of eNOS Signaling in MAEC Overexpressing Prep1

To investigate the role of PREP1 in the endothelium, we transiently transfected Mouse Aortic Endothelial Cells (MAEC) with an empty vector (CTRL) or with *Prep1* cDNA (*Prep1*). The measurement of the PREP1 protein and mRNA levels showed a 3.1- and 25-fold increase, respectively (Figure 1a,b). In parallel, eNOS^Thr495^ phosphorylation was up-regulated 1.5-fold in *Prep1* MAECs compared with CTRL cells, while the amount of phosphorylated eNOS^Ser1177^ and the total eNOS levels did not change (Figure 1c). Next, we analyzed two main activators of eNOS^Thr495^, PKCα, and AMPK. *Prep1* overexpression increased PKCα phosphorylation by 1.6-fold without affecting AMPK. Consistently, the PKC inhibitor, bisindolylmaleimide (BIM), did not modify PREP1 protein and mRNA levels (Figure 2, Appendix A) but reduced PREP1-mediated PKCα and eNOS^Thr495^ phosphorylation by 66% and 58%, respectively (Figure 2). A similar trend was observed in CTRL cells incubated with BIM, despite the fact that this did not reach statistical significance.

### 2.2. Evaluation of Inflammatory and Antioxidant Responses in MAEC Overexpressing Prep1

*Prep1* overexpression increased *Tnf-α* and *Il-6* mRNA levels by 3.5 and 2.3, and this effect was reverted by BIM (Figure 3a,b). In contrast, *Il-1β* was not significantly induced by PREP1 and was not affected by PKC inhibition (Figure 3c). Next, we measured the mRNA levels of the *superoxide dismutase 2* (*Sod2*) and the *activating transcription factor 4* (*Atf4*), as two main genes involved in the antioxidant response. As shown in Figure 3d,e, PREP1 reduced *Sod2* and *Atf4* levels by 50 and 39%, respectively, and consistently with previous data, BIM restored the effect of PREP1. In the control cells, BIM slightly increased mRNA levels of *Atf4* without affecting *Sod2*.

### 2.3. Resveratrol Restores PREP1-Mediated Endothelial Dysfunction

To assess whether resveratrol (RSV) could improve endothelial function, we incubated MAEC overexpressing *Prep1* with 20 μM RSV for 2 h. As shown in Figure 4, RSV reduced PREP1 protein levels by 36 and 70% in CTRL and *Prep1* MAECs, respectively. Moreover, PKCα phosphorylation was decreased by 35- and 37% for RSV, respectively, in CTRL and *Prep1* MAECs, while eNOS^Thr495^ phosphorylation decreased by 27 and 32% (Figure 4). In parallel, in the presence of RSV, *Tnf-α*, and *Il-6* mRNA levels were 52 and 46% lower in the control cells and 82 and 57% lower in *Prep1* MAECs (Figure 5a,b). Interestingly, although not reaching statistical significance, *Il-1β* levels decreased in the presence of RSV (Figure 5c). Finally, RSV increased *Sod2* and *Atf4* mRNA levels by 5- and 3.2-fold in *Prep1* MAECs compared to the *Prep1* untreated cells, respectively (Figure 5d,e), and a similar trend was observed for CTRL cells.

### 2.4. Effect of PREP1 Depletion on Murine Aorta

Next, we investigated the function of PREP1 in the aorta of WT and *Prep1*^i/+^ mice at different ages. WT mice showed a 1.7-fold increase in PREP1 at 18 months compared to 6-month-old mice. By contrast, PREP1 levels were 64 and 50% lower in 6 and 18-month-old *Prep1*^i/+^ mice compared to WT of the same age (Figure 6a). In parallel, PKCα and eNOS^Thr497^ phosphorylation was 54 and 60% lower in *Prep1* hypomorphic mice compared to the WT littermates (Figure 6b). At variance, *Tnf-α* levels were 45% lower in 18-month-old *Prep1*^i/+^ mice compared to the WT littermates (Figure 7a), while *Il-6* did not change and *Il-1β* increased (Figure 7b,c). However, *Sod2* and *Atf4* levels were 48 and 44% lower in *Prep1* deficient compared to WT mice (Figure 7d,e).

## 3. Discussion

TALE (Three Amino Acid Loop Extension) proteins are a group of homeodomain transcription factors (TFs) that are characterized by the presence of a loop formed by three amino acids that extend between the first and second α-helix of the homeodomain [22,23,24]. Among these TFs, PREP1 plays an important role in the regulation of glucose and lipid homeostasis [15,17,18,19]. Indeed, *Prep1^i/+^* mice are protected from streptozotocin-induced diabetes and feature reduced lipotoxicity and diet-induced steatohepatitis [18,19]. Previous studies performed in MAECs indicated that PREP1 stimulates the formation of new blood vessels by increasing angiogenic molecules, some of which are also involved in vascular dysfunction during aging or metabolic diseases [20]. More recently, we have shown how PREP1 increases in the aorta of aged mice and in senescent vascular smooth muscle cells (VSMCs) and is involved in the neointimal formation of reducing VSMCs apoptosis. In particular, PREP1 mediates Il-6-antiapoptotic action on VSMCs through a mechanism involving both p53 and Bcl-xL, and this effect is particularly clear in senescence [21].

To date, it is clear that with advancing age or in the case of metabolic imbalance, such as that occurring in type 2 diabetes, blood vessels not only tend to thicken but also show a marked endothelial dysfunction. This pathological condition can favor the pathogenesis of atherosclerosis, and in patients with high cardiovascular risk, a reduced endothelium-dependent vasodilatation is a predictive indicator of cardiovascular events. At the molecular level, several mechanisms have been described to impair endothelial function, including reactive oxygen species (ROS) and inflammatory cytokine production, including a reduction in the eNOS pathway. In particular, the enzymatic activity of eNOS is regulated by phosphorylation at multiple sites, including serine 1177 (Ser1177) and threonine 495 (Thr495). The Ser1177 site is rapidly phosphorylated in response to some hormones and induces an increase in the enzyme’s activity resulting in the synthesis of NO. Conversely, the Thr495 site is constitutively phosphorylated and results in the inhibition of enzyme activity [25,26].

Based on these premises, in the current work, we have examined the role of PREP1 on endothelial dysfunction during aging.

The *Prep1* overexpression in MAECs increases eNOS^Thr495^ without affecting the total levels and the phosphorylation on Ser^1177^ of eNOS. eNOS^thr495^ phosphorylation can be induced by several protein kinases, including AMPK and protein kinases C (PKCs) [5,6,7,8,9]. However, AMPK phosphorylation was similar in the control and *Prep1* MAEC, while classical PKCα was significantly induced by PREP1. In parallel, bisindolylmaleimide (BIM), as a pharmacological PKC inhibitor, reverted the effect of PREP1 on eNOS^thr495^. Although we cannot exclude the involvement of other PKCs on eNOS^thr495^ phosphorylation, previous studies that were performed using activators or inhibitors of the classical PKC have driven our focus toward PKCα isoform [7,8]. Interestingly, PREP1 enhanced the expression of proinflammatory cytokines, *Tnf-α* and *Il-6*, and decreased *Sod2* and *Atf4*: two main molecules involved in the antioxidant response. The inhibition of PKCα by BIM reverted the effect of PREP1, confirming the possible role of PKCα on PREP1-mediated endothelial dysfunction. In addition, in the control cells, BIM significantly increased *Atf4* mRNA levels; a possible explanation for this result is that PKCs could also modulate Atf4 expression independently on PREP1. Indeed, in a manuscript by Woo S. et al., it was shown that the expression of *Atf4* increased when PKC was inhibited in mouse auditory HEI-OCI cells [27].

Several studies have shown that diet-derived polyphenols can be protective factors against cardiovascular and metabolic diseases or biological aging. In particular, the positive effects of resveratrol (RSV), as a polyphenol found in grapes and red wine, have been largely studied and associated with a reduction in inflammatory markers and ROS production [28,29]. In parallel, several studies have highlighted the antiaging properties of RSV as well as its beneficial effects on CVD. However, further evidence is needed to confirm its effectiveness [30,31,32,33,34,35]. At the molecular level, RSV displays antioxidant and anti-inflammatory action in numerous organs and tissues, including the cardiovascular system [28,36,37,38]. Furthermore, this polyphenol inhibits PKCα and stimulates the production of nitric oxide through the activation of eNOS, which enhances the integrity of the endothelium [39,40]. Thus, to evaluate the possible role of RSV on the PREP1-mediated endothelial dysfunction, MAEC cells overexpressing *Prep1* were incubated with this compound. RSV significantly reduced PREP1 expression and inhibited PREP1-mediated PKCα/eNOS^Thr495^ phosphorylation. In parallel, RSV decreased *Tnf-α* and *Il-6* induced by PREP1 and increased *Sod2* and *Atf4* expression (Figure 8). Actually, whether RSV can regulate PREP1 levels directly or indirectly through the modulation of some PREP1 inducers is not clear. We have shown that Il-6 can increase *Prep1* expression in VSMC [21]. In addition, Whung B.S. et al. found RSV to inhibit Il-6 in endothelial cells at the promoter, transcriptional, and protein levels [34]. Thus, we could suppose this interleukin to be a mediator of RSV on PREP1 expression, and this hypothesis could be considered the aim of further studies.

In order to investigate the role of PREP1 in vivo*,* we performed experiments on aged *Prep1^i/+^* mice, which expressed low levels of this protein. This animal model is quite interesting because, as described above, it has better insulin sensitivity and reduced neointimal formation [17,18,19,21]. In addition, since PREP1 increases during WT mice aging [21], it is possible to hypothesize the possible role of this transcription factor in age-related diseases. Our results indicate that PREP1 protein levels were significantly reduced in the total aortic tissue from aged *Prep1^i/+^* mice compared to age-matched WT mice. This result is consistent with the *Prep1* mRNA levels previously measured and described in Cimmino et al. [21]. In addition, PKCα and eNOS^thr495^phosphorylation and the *Tnf-α* mRNA content were significantly reduced, while *Il-6* did not change in *Prep1^i/+^* mice compared to WT mice. Conversely, *Il-1β* increased in *Prep1*-deficient mice, and this result could depend on the tunica media or tunica adventitia and not directly on the endothelium. Finally, the expression of *Sod2* and *Atf4* were downregulated in *Prep1^i/+^* mice, despite their lower *Prep1* levels. These data suggest that in vivo, the decrease in *Prep1* alone restored eNOS activation and, in part, inflammatory response; however, it was not sufficient enough to improve age-induced oxidative stress. A possible explanation could be that since aged *Prep1^i/+^* mice already have the increased activation of the eNOS pathway, they do not need a reduction in oxidative stress.

Current data provide new knowledge about the role of PREP1 in the aged aorta, indicating that this transcription factor not only acts in the medial muscle component of the aortic wall but also at the endothelial level. In addition, resveratrol, at least in vitro, decreases PREP1 and improves endothelial function, suggesting this transcription factor to be a possible pharmacological target in metabolic and aging-related vascular dysfunction.

To date, it is clear that the restoration of the endothelial function could represent a therapeutic target, especially considering that some antihypertensive or antidiabetic drugs or statins can improve endothelium-dependent vasodilatation [41]. However, the results of pharmacological studies on the endothelial function have produced conflicting results, which depend on the applied stimulus and the explored vascular district. Indeed, a class of drugs can partially improve or restore the normal endothelial function in this district, while it may not be effective in another. In addition, drugs can act on distinct endothelial signaling pathways [42]. For this reason, the discovery of PREP1 as a new target of resveratrol could have clinical relevance. However, as several authors have shown that resveratrol improves metabolism and endothelial dysfunction in mice [43,44,45], a limitation of this study, which needs to be further investigated, is the lack of the in vivo effect of resveratrol on endothelial function in *Prep1^i/+^* mice.

## 4. Materials and Methods

### 4.1. Materials

Media, sera, antibiotics for cell cultures, and the Lipofectamine 2000 Trasfection Reagent (Catalog#: 11668019, Invitrogen, Grand Island, NY, USA) were all from Invitrogen (Grand Island, NY, USA). *Prep1* plasmid cDNA (pRc/CMV-*Prep1*) was designed in the laboratory and produced by Invitrogen (Grand Island, NY, USA).

Protein electrophoresis and real-time PCR reagents were from Bio-Rad (Richmond, VA, USA). Western blotting and ECL reagents were purchased from Amersham Biosciences (Arlington Heights, IL, USA) and EuroClone (EuroClone SpA, Milan, Italy).

### 4.2. Cell Culture Procedures, Transient Transfection, and Cell Proliferation

Mouse Aortic Endothelial Cells (MAEC) were gifts from Prof. Dr. Angelika Bierhaus from the University of Heidelberg, Germany. The cells were plated in T75 flasks and grown in DMEM containing 1 g/L of glucose supplemented with 10% of the Fetal Bovine Serum (FBS), 2 mmol/L L-glutamine, 0.1 mmol/L non-essential amino acids, and 1% penicillin/streptomycin solution at 37 °C in a humidified atmosphere, 95% air and 5% CO_2_ (all *v*/*v*). The transient transfection of *Prep1* plasmid cDNA (pRc/CMV*Prep1*) in MAEC cells was performed using the Lipofectamine reagent (Invitrogen). Cells (200 × 10^5^) were grown in 6-well plates to about 60% confluence. After overnight starvation, the cells were incubated (5 h, 37 °C) with serum-free DMEM containing plasmid DNA (2 µg/well) and lipofectamine in a 1:3 proportion. Then, a fresh medium supplemented with FBS was added to the plate to reach a 10% concentration. Western blot and qPCR were performed after each transfection to assess transfection efficiency prior to the downstream experiments.

### 4.3. Western Blot Analysis

Cells were solubilized with a lysis buffer containing 50 mM HEPES, 150 mM NaCl, 10 mM EDTA, 10 mM Na_4_P_2_O_7_, 2 mM sodium orthovanadate, 50 mM NaF, 1 mM phenylmethylsulfonyl fluoride, 10 μg/mL aprotinin, 10 μg/mL leupeptin, pH 7.4, and 1% (*v*/*v*) Triton X-100. Lysates were clarified by centrifugation at 12,000× *g* for 20 min at 4 °C. The protein concentrations in the cell lysates were measured at a Beckman Coulter spectrophotometer (DU^®^530, Life Science UV/VIS) using the Breadford Assay by Bio-Rad DC (detergent compatible, Bio-Rad Protein Assay Dye Reagent Concentrate, Catalog#: 5000006, Bio-Rad, Hercules, CA, USA). Proteins (50 µg) were resolved by dodecyl sulfate-polyacrylamide gel electrophoresis (SDS-PAGE), transferred onto the PVDF membrane (Amersham Hybond P0.45 PVDF, Catalog#: 10600023, Amersham Biosciences, Arlington Heights, IL, USA), and blocked with 5% BSA in Tris-buffered saline containing tween 20. Membranes were incubated with the specific primary antibodies: anti-PREP1 (1:500, Catalog#: sc-25282, Santa Cruz Biotechnology, Dallas, TX, USA); anti-peNOS^ser1177^ (1:1000, Catalog#: 9571, Cell Signalling technology, Danvers, MA, USA); anti-peNOS^Thr495^ (1:1000, Catalog#: 9574, Cell Signalling technology, Danvers, MA, USA); anti-eNOS (1:1000, Catalog #: 9572, Cell Signalling technology, Danvers, MA, USA); anti-pPKCα (1:500, Catalog #: 06-822, MilliporeSigma, Burligton, MA, USA); anti-PKCα (1:1000, Catalog#: sc-8393, Santa Cruz Biotechnology, Dallas, TX, USA); anti-AMPK (1:1000; Catalog#: 07-350, MilliporeSigma, Burligton, MA, USA); anti-pAMPK (1:1000; Catalog#: 2531S, Cell Signalling technology, Danvers, MA, USA); anti-ACTIN (1:1000, Catalog#: sc-47778, Santa Cruz Biotechnology, Dallas, TX, USA). Secondary antibodies were all diluted 1:1000 and were anti-rabbit (Catalog#: 170-6515, Bio-Rad, Hercules, CA, USA), anti-mouse (Catalog#: 170-6516, Bio-Rad, Hercules, CA, USA), and anti-goat (Catalog#: sc-2020, Santa Cruz Biotechnology, Dallas, TX, USA). Antigen-antibody complexes were revealed by enhanced chemiluminescent detection kit (LiteAblot^®^PLUS, EuroClone SpA, Milan, Italy) according to the manufacturer’s instruction. Densitometric analyses have been performed on autoradiographs with Image Lab software (Bio-Rad, Hercules, CA, USA).

### 4.4. Real-Time RT-PCR Analysis

The total cellular RNA was isolated from MAEC cells using the RNeasy kit (QIAGEN Sciences, Hilden, Germany) according to the manufacturer’s instructions. In total, 1 µg of cell RNA was reverse-transcribed using Superscript III Reverse Transcriptase (Life Technologies Carlsbad, CA, USA). PCR reactions were analyzed using IQTM SYBR Green Supermix (Bio-Rad, Hercules, CA, USA). The reactions were performed using Platinum SYBR Green qPCR Super-UDG and using an iCycler IQ Multicolor Real-Time PCR Detection System (Biorad, CA, USA), running a total of 40 cycles. All reactions were performed in triplicate, and expression data were normalized to the geometric mean of the housekeeping gene beta-actin, which was analyzed using the 2^−ΔCT^ or 2^−ΔΔCT^ methods [46]. The primer sequences used are described in electronic Appendix A.

### 4.5. In Vivo Experiments

*Prep1* targeted mice were generated by gene trapping through Lexikon Genetics, Inc. (The Woodlands, TX, USA). C57BL/6J *Prep1^i/+^* male mice (*Prep1^i/+^*) and C57BL/6J wildtype (WT) male littermates were housed in two to three per cage at a constant temperature and relative humidity and were acclimated to a 12 h light/dark cycle and had ad libitum access to food and water. Their general phenotype has been previously described [19]. All animal handling was approved by the Ethics Committee on Animal Use of the University of Naples “Federico II” (permission number 176/2020-PR, prot. 39F3A.60) and complied with the standards of the European Union. WT and *Prep1^i/+^* mice 6- and 18-month-old were euthanized, and the aorta was extracted, snap-frozen in liquid nitrogen, and stored at −80 °C for subsequent use.

### 4.6. Statistical Procedures

Data were analyzed with GraphPad Prism 8.0 (GraphPad Inc., San Diego, CA, USA). For comparisons between the two groups, a two-tailed *t*-test for independent samples was used. Multiple comparisons among more groups were made using the ANOVA test with Tukey’s correction (for normally distributed data). *p* values equal to or less than 0.05 were considered statistically significant.

## Figures and Tables

**Figure 1 ijms-24-11891-f001:**
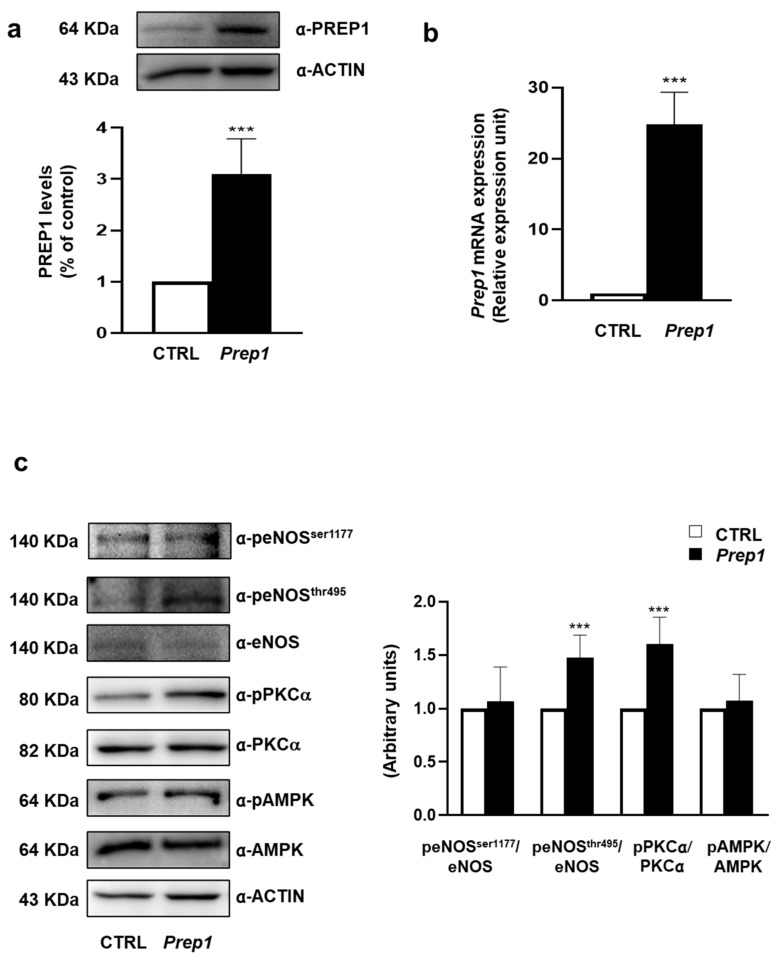
*Prep1* overexpression in MAEC and eNOS signaling. (**a**,**c**) Mouse Aortic Endothelial Cells (MAEC) were transfected with *Prep1* cDNA and an empty vector (CTRL). Protein levels were analyzed by Western blot using antibodies for PREP1, peNOS^Ser1177^, peNOS^Thr495^, eNOS, pPKCα, PKCα, AMPK, AMPK, and ACTIN, which was used for normalization. The autoradiographs shown are representative of three different experiments that were subjected to densitometric analysis. (**b**) *Prep1* levels were analyzed by real-time RT-PCR analysis, using beta-actin as the internal standard. Bars represent the mean ± SD of three independent experiments, each performed in triplicate. Asterisks denote statistical differences (*** *p* < 0.001).

**Figure 2 ijms-24-11891-f002:**
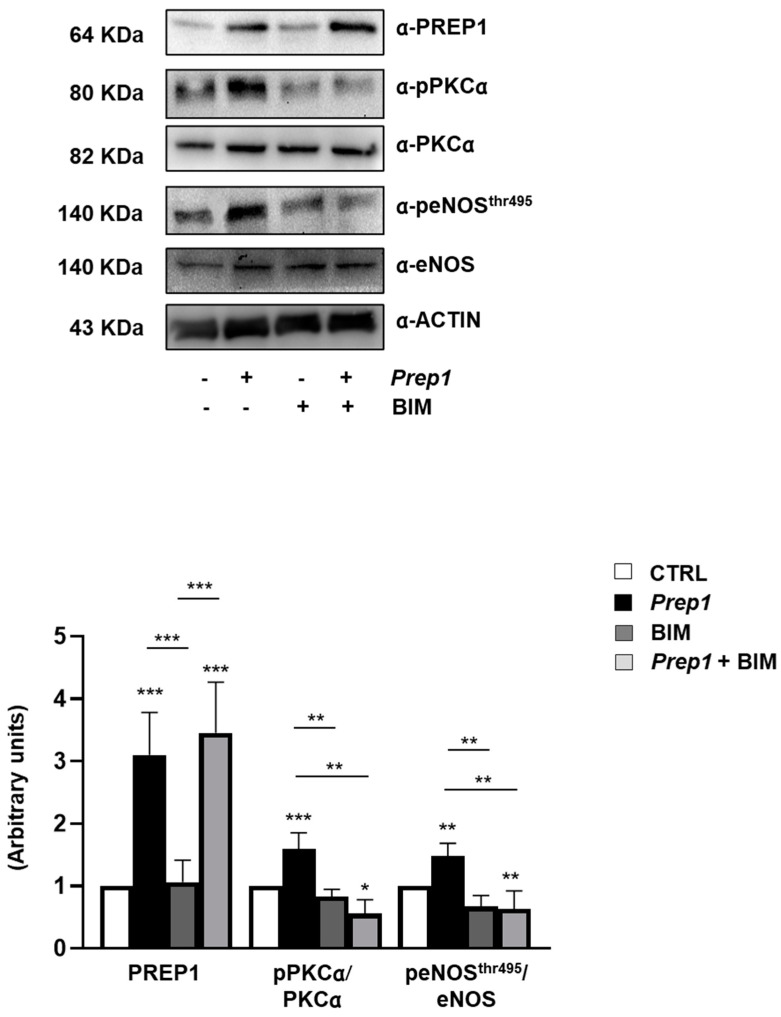
The role of PKCα on PREP1-mediated eNOS signaling. CTRL and *Prep1* overexpressed MAEC cells were incubated with bisindolylmaleimide (BIM) at a 100 nM concentration for 30 min. The protein expression of PREP1, pPKCα, PKCα, peNOS^Thr495^, and eNOS was analyzed by Western Blot using the ACTIN antibody for normalization. PKCα and eNOS bands were taken from a parallel gel loaded with the same lysates. The autoradiographs shown are representative of three different experiments and subjected to densitometric analysis. Bars represent the mean ± SD of three independent experiments, each performed in triplicate. Asterisks denote statistical differences (* *p* < 0.05; ** *p* < 0.01; *** *p* < 0.001).

**Figure 3 ijms-24-11891-f003:**
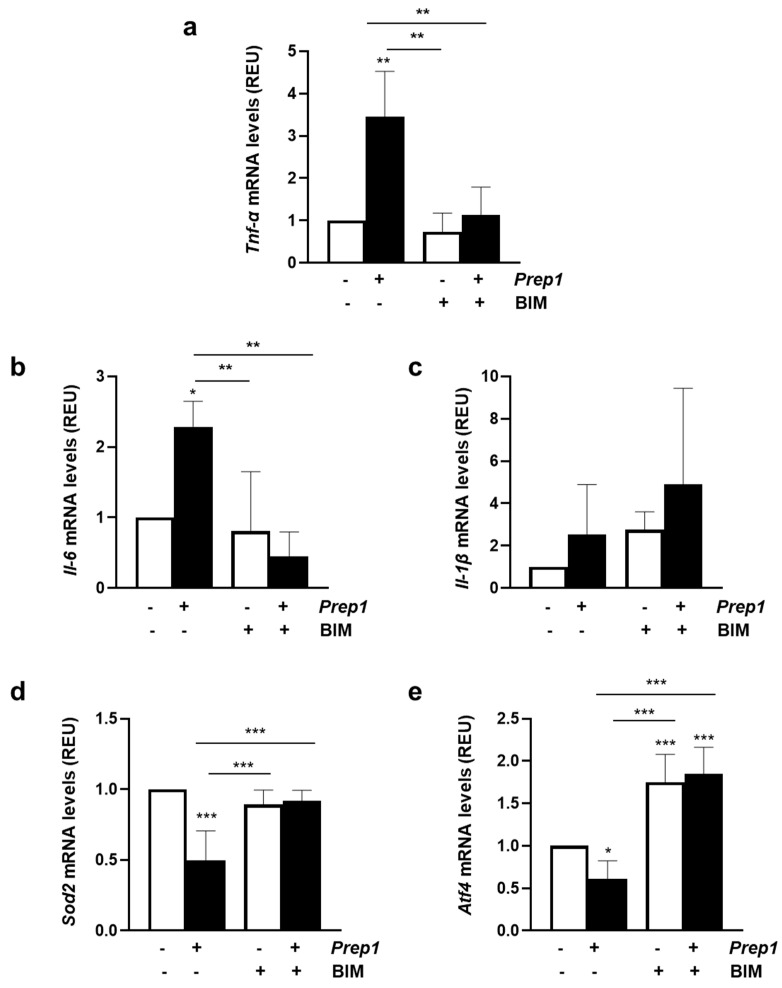
The role of PKCα on PREP1-mediated expression of proinflammatory cytokines and antioxidant molecules. CTRL and *Prep1* overexpressed in MAEC cells were incubated with bisindolylmaleimide (BIM) at a 100 nM concentration for 30 min. *Tnf-α* (**a**), *Il-6* (**b**), *Il-1β* (**c**), *Sod2* (**d**), and *Atf4* (**e**) mRNA levels were evaluated by real-time RT-PCR analysis in CTRL and *Prep1* overexpressing MAEC cells. Data were normalized by the amount of *beta-actin* used as an internal control. Bars represent the mean ± SD of three independent experiments, each performed in triplicate. Asterisks denote statistical differences (* *p* < 0.05; ** *p* < 0.01; *** *p* < 0.001).

**Figure 4 ijms-24-11891-f004:**
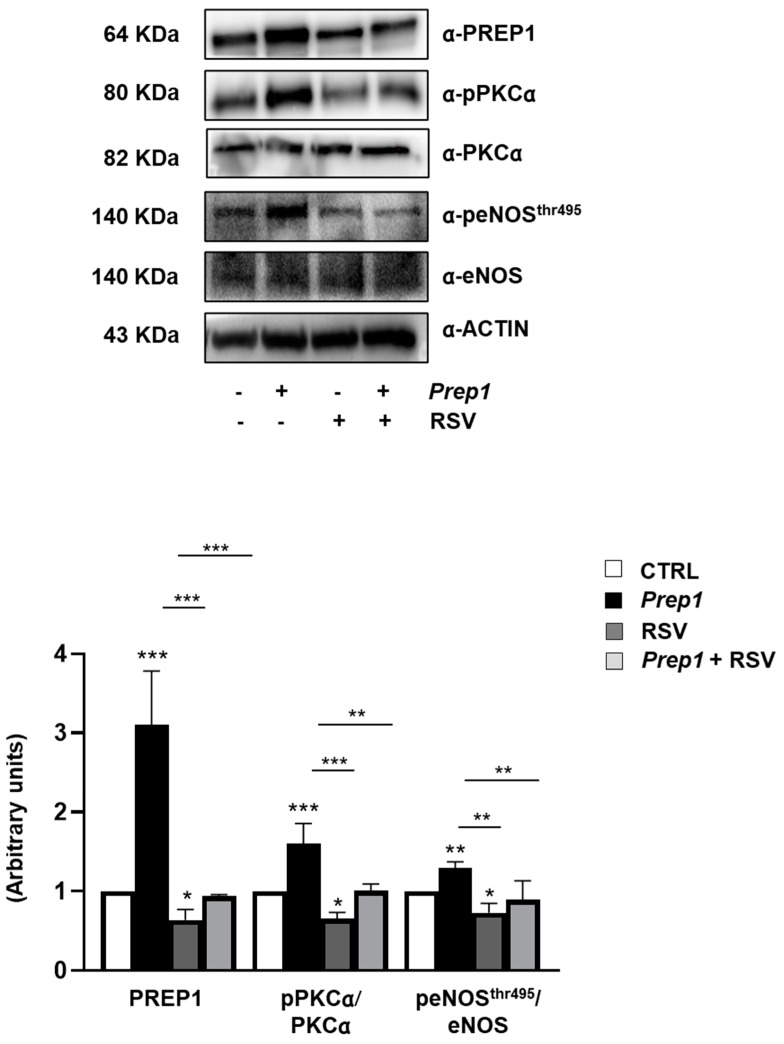
The role of resveratrol on PREP1-mediated eNOS signaling. CTRL and *Prep1* overexpressed in MAEC cells were incubated with resveratrol (RSV) at a 20 μM concentration for 2 h. The protein expression of PREP1, pPKCα, PKCα, peNOS^Thr495^, and eNOS was analyzed by Western Blot using ACTIN antibody for normalization. The autoradiographs shown are representative of three different experiments and subjected to densitometric analysis. Bars represent the mean ± SD of three independent experiments, each performed in triplicate. Asterisks denote statistical differences (* *p* < 0.05; ** *p* < 0.01; *** *p* < 0.001).

**Figure 5 ijms-24-11891-f005:**
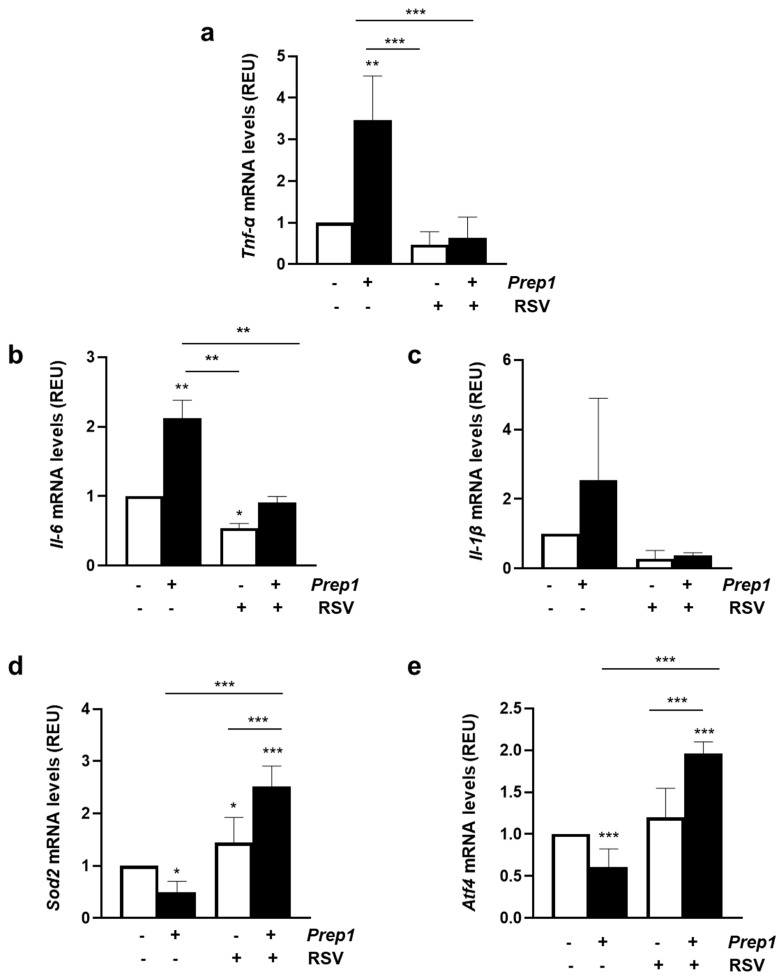
The role of resveratrol on the PREP1-mediated expression of proinflammatory cytokines and antioxidant molecules. CTRL and *Prep1* overexpressed in MAEC cells were incubated with resveratrol (RSV) at 20 μM concentration for 2 h. *Tnf-α* (**a**), *Il-6* (**b**), *Il-1β* (**c**), *Sod2* (**d**), and *Atf4* (**e**) mRNA levels were evaluated by a real-time RT-PCR analysis in CTRL, and *Prep1* was overexpressed MAEC cells. Data were normalized by the amount of *beta-actin* used as an internal control. Bars represent the mean ± SD of three independent experiments, each performed in triplicate. Asterisks denote statistical differences (* *p* < 0.05; ** *p* < 0.01; *** *p* < 0.001).

**Figure 6 ijms-24-11891-f006:**
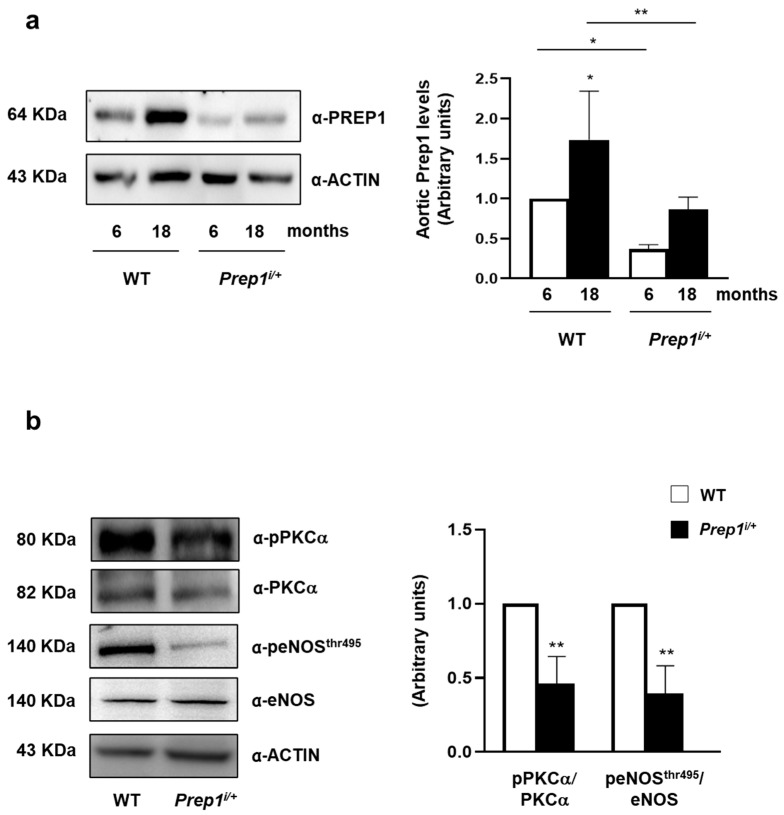
The role of PREP1 on eNOS signaling in WT and *Prep1^i/+^* mice. (**a**,**b**) Aortas from 6- to 18-month-old WT and *Prep1^i/+^* mice were solubilized, and protein samples were analyzed by a Western Blot with PREP1, pPKCα, PKCα, peNOS^Thr495^, eNOS antibodies. ACTIN antibody was used for normalization. Blots were revealed by ECL, and the autoradiograph was representative of three independent experiments and subjected to densitometric analysis. eNOS bands were taken from a parallel gel loaded with the same lysates. Asterisks denote statistical differences (* *p* < 0.05; ** *p* < 0.01). Arbitrary units are related to fold changes relative to WT.

**Figure 7 ijms-24-11891-f007:**
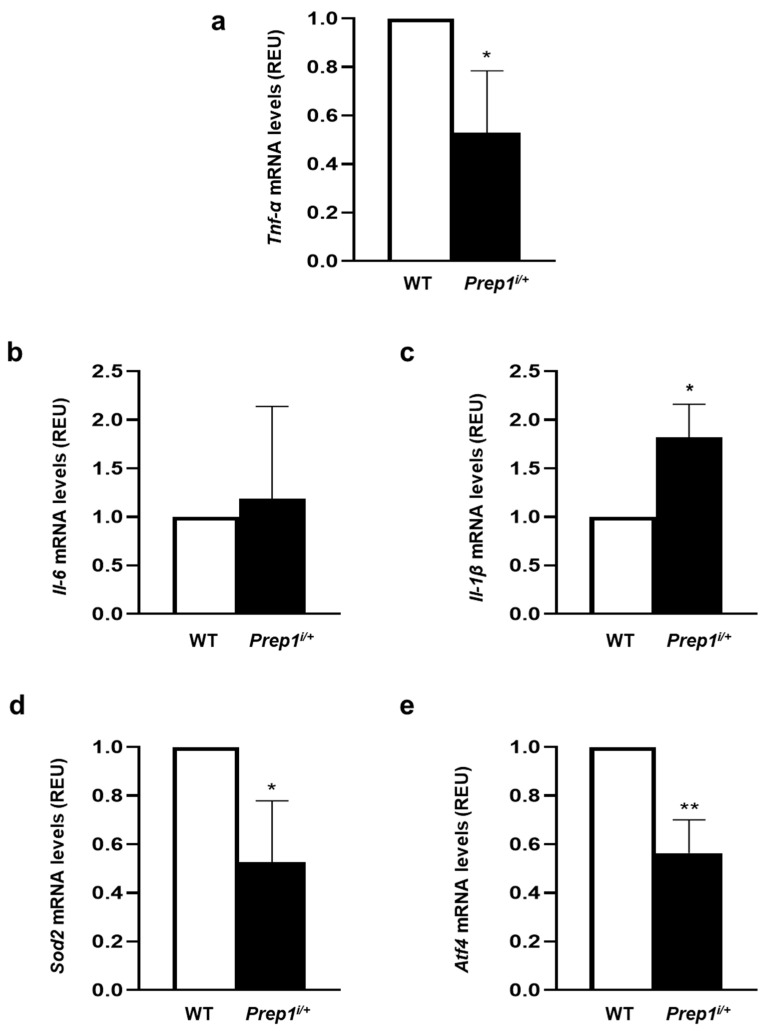
The role of PREP1 on proinflammatory cytokines and antioxidant gene expression in WT and *Prep1^i/+^* mice. *Tnf-α* (**a**), *Il-6* (**b**), *Il-1β* (**c**), *Sod2* (**d**), and *Atf4* (**e**) mRNAs were determined by a real-time RT-PCR analysis of the total RNA isolated from the aorta of 18-month-old WT (*n* = 5) and *Prep1^i/+^* (*n* = 5) mice, using *β-actin* as an internal standard. Bar represents the mean ± SD of three independent experiments, the reactions of which were each performed in triplicate using the total RNAs obtained from six mice per genotype. Asterisks denote statistical differences (* *p* < 0.05; ** *p* < 0.01).

**Figure 8 ijms-24-11891-f008:**
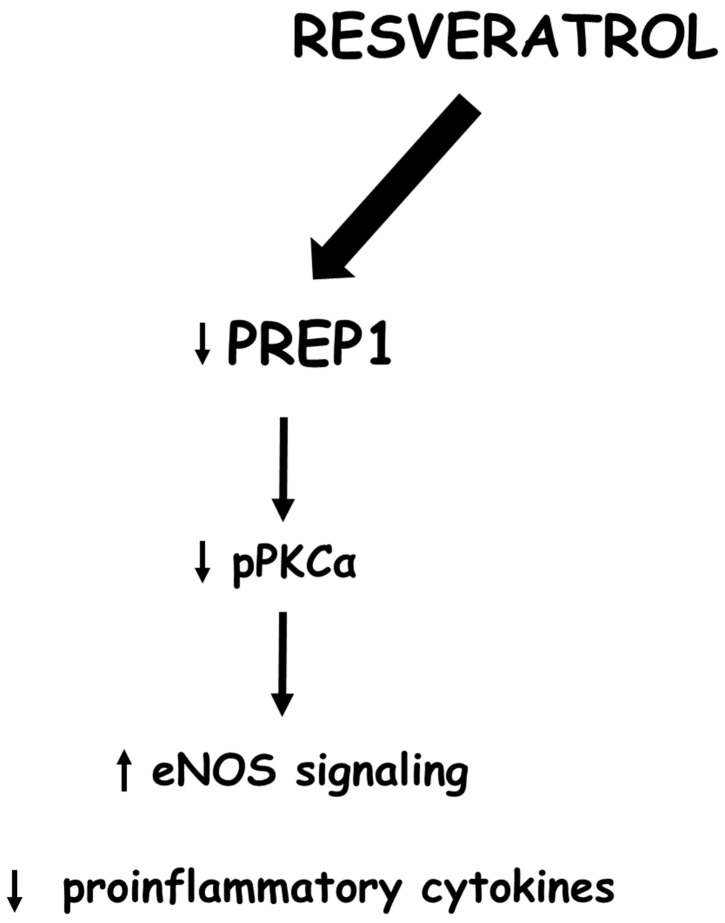
Schematic representation of resveratrol action on PREP1 signaling: Resveratrol, down-regulating PREP1, reduces PKCα and eNOS^Thr495^ phosphorylation and proinflammatory cytokine expression. In addition, resveratrol increases the levels of antioxidant molecules, suggesting PREP1 to be a new pharmacological target of this compound.

## Data Availability

The data that support the findings of this study are available upon reasonable request.

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
