# Peer review of "Resveratrol Improves Endothelial Function by A PREP1-Mediated Pathway in Mouse Aortic Endothelial Cells"

_ijms, 2023, doi:10.3390/ijms241511891_

Round 1

Reviewer 1 Report

Following are my comments for the manuscript:

1) Nicely written abstract and introduction, however, first paragraph of introduction can be more elaborated and expanded; there is a language error on line 67 "performed"

2) Language error on line 75..it should read as measurement instead of measure

3) In Figure 1C, western blot for total protein eNOS does not even have bands; not convincing

4) Please explain results of figure 3E mentioned in line 109; why BIM increased levels of ATF4 in controls also?

5) Please comment on figure 5C; is bigger error bars are technical replicate error as RSV treated samples looks significant visually

6) In figure 6, are the arbitrary units related to fold changes relative to WT?

7) Please explain results of figure 7C with respect to increase in IL-1B in Prep1i/+ mice?

8) There is a sentence error in line 216; it will be better to mention anti-inflammatory genes before SOD2 and ATF4

9) Overall very nicely written discussion covering previous work flows, results from current work and its impact on rationalizing future studies; any comments on in vivo treatment of Resveratrol to strengthen the in vitro phenotype?

Author Response

Reviewer #1:

  1. Nicely written abstract and introduction, however, first paragraph of introduction can be more elaborated and expanded; there is a language error on line 67 "performed”.

Answer:

We thank the Reviewer for this suggestion. Introduction paragraph has been expanded exploiting the role of different transcription factors in the action of resveratrol (P1/L45).

As requested by the reviewer, we have changed the word “performed” with “showed” (P2/L84).

  1. Language error on line 75.it should read as measurement instead of measure

Answer:

As requested by the reviewer, we have changed the word “measure” with “measurement” (P2/L92).

  1. In Figure 1C, western blot for total protein eNOS does not even have bands; not convincing.

Answer:

We thank the Reviewer for this suggestion. We have now shown a better exposure of the eNOS blot in Figure 1C. We think it should now be clearer.

  1. Please explain results of figure 3E mentioned in line 109; why BIM increased levels of ATF4 in controls also?

Answer:

We thank the Reviewer for raising this interesting point. A possible explanation about the increased levels of ATF4 in control MAECs is that PKCs may modulate ATF4 expression also independently from Prep1. In a manuscript by Seon Min Woo et al., it is shown that the expression of ATF4 increases when PKC is inhibited in mouse auditory HEI-OCI cells. This important point has been addressed in discussion (P10/L231).

  • Seon Min Woo et al. Mol Med Rep. 2016 Jul;14(1):845-50. doi: 10.3892/mmr.2016.5320.

  1. Please comment on figure 5C; is bigger error bars are technical replicate error as RSV treated samples looks significant visually

Answer:

We agree with the Reviewer that IL-1β has a tendency to increase. However, as IL-1β is expressed at low levels in our cells, the small changes may account for a very high variability.

  1. In figure 6, are the arbitrary units related to fold changes relative to WT?

Answer:

Yes, they are. This important point has been added in figure 6 legend (P8/L182).

  1. Please explain results of figure 7C with respect to increase in IL-1B in Prep1i/+ mice?

Answer:

We thank the reviewer for raising this point. Experiments in mice have been performed in the aortic wall. Although we do not have a definite answer, we may suppose that the increased levels of IL-1β could depend on the tunica media or tunica adventitia and not directly on the endothelium. This important point has been addressed in discussion (P11/L266).

  1. There is a sentence error in line 216; it will be better to mention anti-inflammatory genes before SOD2 and ATF4

Answer:

Done as requested (P10/L249).

  1. Overall very nicely written discussion covering previous workflows, results from current work and its impact on rationalizing future studies; any comments on in vivo treatment of Resveratrol to strengthen the in vitro phenotype?

Answer:

We thank the reviewer for raising this point. Several authors have suggested a positive effect of resveratrol on metabolism and endothelial dysfunction in mice. Unfortunately, we still do not have data about the effect of resveratrol on endothelial function in Prep1i/+ mice and this is a limit of the present manuscript, as now reported in discussion (P11/L279).

Reviewer 2 Report

Cabaro et al. investigated in this article the effect of resveratrol on mouse endothelial function by evaluation the Prep1 transcription factor pathway.

Resukts show that overexpression of Prp1 cDNA in aortic endothelial cells increases the level pf Prep1 mRNA and protein concomitant to the increase in the phosphorylation of eNOS -and PKC-a without change in AMPK status. The role of PKCa in eNOS phosphorylation was attested by PKCa inhibitor, which seems to be involved in Prep1 overexpression-associated induction of proinflammatory cytokines Tnfa and Il-6 mRNAs and a decrease in Sod2 and ATF4 mRNAs. Treatment of MAEC cells with resveratrol abrogates the effects of Prep1 overexpression. On the other hand, 18 months-aged hypomorphic heterozygous (Prep1i/+) mouse models expressing approximatively fifty percent of Prep1 protein, show different response than in MACE cells, while the effect of resveratrol in vivo has not been investigated.   

Major points:

-       If the rational is to investigate the improvement of endothelial function by resveratrol, it’s in vivo effect should be reported.

-       It’s quite difficult to compare the original blots to the blot reported in the article. Authors didn’t show the entire blots with the molecular weight markers. In each blot figure legend, they indicate : “Bars represent the mean ± SD of three independent experiments”. This is not clear on the original blots ? Elsewhere, for example, Fig 2 BIM, it seems that the reported PKC and pPKC, and eNOS and peNOS are issued from different blots?

Minor points:

-       Authors should use the common guidelines for nomenclature for genes and proteins: (i) Not italicized protein symbols use all uppercase letters (PREP1, ACTIN…etc); and (ii) italicized gene name begins with an uppercase letter (not a number), followed by all lowercase letters (Tnfa, Il-6…etc).

-       For all western blotting bands, the molecular weight of each band should be indicated.

Material and methods:

-       For transfection experiments: indicate the number of cells, plates, quantity of Prep1plasmid used and ratio to lipofectamine. How transfection efficiency was standardized from one well to another?

-       Antibody dilutions?

-       qPCR cycling?

-       Mice gender?

Author Response

Reviewer #2: 

Cabaro et al. investigated in this article the effect of resveratrol on mouse endothelial function by evaluation the Prep1 transcription factor pathway.

Results show that overexpression of Prp1 cDNA in aortic endothelial cells increases the level pf Prep1 mRNA and protein concomitant to the increase in the phosphorylation of eNOS -and PKC-a without change in AMPK status. The role of PKCa in eNOS phosphorylation was attested by PKCa inhibitor, which seems to be involved in Prep1 overexpression-associated induction of proinflammatory cytokines Tnfa and Il-6 mRNAs and a decrease in Sod2 and ATF4 mRNAs. Treatment of MAEC cells with resveratrol abrogates the effects of Prep1 overexpression. On the other hand, 18 months-aged hypomorphic heterozygous (Prep1i/+) mouse models expressing approximatively fifty percent of Prep1 protein, show different response than in MACE cells, while the effect of resveratrol in vivo has not been investigated.  

Major comments:

  1. If the rational is to investigate the improvement of endothelial function by resveratrol, it’s in vivo effect should be reported.

Answer:

We thank the reviewer for raising this point. Several authors have suggested a positive effect of resveratrol on metabolism and endothelial dysfunction in mice. Unfortunately, we still do not have data about the effect of resveratrol on endothelial function in Prep1i/+ mice and this is a limit of the present manuscript, as now reported in discussion (P11/L279). However, we have changed the title as follows “Resveratrol improves endothelial function by a Prep1-mediated pathway in mouse aortic endothelial cells”.

  1. It’s quite difficult to compare the original blots to the blot reported in the article. Authors didn’t show the entire blots with the molecular weight markers. In each blot figure legend, they indicate: “Bars represent the mean ± SD of three independent experiments”. This is not clear on the original blots? Elsewhere, for example, Fig 2 BIM, it seems that the reported PKC and pPKC, and eNOS and peNOS are issued from different blots?

Answer:

We thank the Reviewer for raising this point. Most of the molecular weight of the blots shown in the full gel scan are visible on the autoradiographs. However, in order to make them clearer, we have now shown a better exposure of some images and added next to the original figure in the full gel scan.

Unfortunately, in figure 2, since eNOS and PKCα hybridization on the same blot was of very poor quality and not visible, we loaded the same lysates on a parallel gel. We have added this information in figure 2 legend (P4/L116).

Minor comments: 

  1. Authors should use the common guidelines for nomenclature for genes and proteins: (i) Not italicized protein symbols use all uppercase letters (PREP1, ACTIN…etc); and (ii) italicized gene name begins with an uppercase letter (not a number), followed by all lowercase letters (Tnfa, Il-6…etc).

Answer:

Done as requested.

  1. For all western blotting bands, the molecular weight of each band should be indicated.

Answer:

We thank the Reviewer for this suggestion. Molecular weight markers have been added to all the blots.

  1. For transfection experiments: indicate the number of cells, plates, quantity of Prep1plasmid used and ratio to lipofectamine. How transfection efficiency was standardized from one well to another?

Answer:

We have better described the transfection protocol and added this information in material and methods (P13/L312).

  1. Antibody dilutions?

Answer:

We have added the antibody dilutions in material and methods (P13/L327).

  1. qPCR cycling?

Answer:

Experiments were intended up to 40 cycles. We have added this information in material and methods (P13/L346).

  1. Mice gender?

Answer:

All mice were male. We have added this information in material and methods (P13/L353).

Round 2

Reviewer 2 Report

One nomenclature point : I have suggested to the authors should use the common guidelines for nomenclature for genes and proteins: (i) Not italicized protein symbols use all uppercase letters (PREP1, ACTIN…etc); and (ii) italicized gene name begins with an uppercase letter (not a number), followed by all lowercase letters (Tnfa, Il-6…etc).

for mRNA it was done, but not for proteins. In all Wblots figures (a-ACTIN instead of a-actin, PREP1 instead of Prep1...etc) and related histograms.

Also in line 40, 78, and all Prep1 when you talk about the protein!

Author Response

Minor comments: 

One nomenclature point: I have suggested to the authors should use the common guidelines for nomenclature for genes and proteins: (i) Not italicized protein symbols use all uppercase letters (PREP1, ACTIN…etc); and (ii) italicized gene name begins with an uppercase letter (not a number), followed by all lowercase letters (Tnfa, Il-6…etc).

for mRNA it was done, but not for proteins. In all Wblots figures (a-ACTIN instead of a-actin, PREP1 instead of Prep1...etc) and related histograms.

Also in line 40, 78, and all Prep1 when you talk about the protein!

Answer:

Done as requested.